# Deciphering Nicotine-Driven Oncogenesis in Head and Neck Cancer: Integrative Transcriptomics and Drug Repurposing Insights

**DOI:** 10.3390/cancers17091430

**Published:** 2025-04-24

**Authors:** Guo-Rung You, Daniel Yu Chang, Hung-Han Huang, Yin-Ju Chen, Joseph T. Chang, Ann-Joy Cheng

**Affiliations:** 1Department of Medical Biotechnology and Laboratory Science, College of Medicine, Chang Gung University, Taoyuan 33302, Taiwan; d000017007@cgu.edu.tw (G.-R.Y.); daniel.chang0116@gmail.com (D.Y.C.); d1001402@cgu.edu.tw (H.-H.H.); 2Graduate Institute of Biomedical Sciences, College of Medicine, Chang Gung University, Taoyuan 33302, Taiwan; yinjuchen@mail.cgu.edu.tw; 3Department of Biomedical Sciences, College of Medicine, Chang Gung University, Taoyuan 33302, Taiwan; 4Department of Radiation Oncology and Proton Therapy Center, Linkou Chang Gung Memorial Hospital, Taoyuan 333423, Taiwan; jtchang@cgmh.org.tw; 5School of Medicine, Chang Gung University, Taoyuan 33302, Taiwan

**Keywords:** head and neck cancer (HNC), nicotine, transcriptomic analysis, drug repurposing, oncogenic pathway

## Abstract

HNC cell lines exposed to nicotine for three months exhibited increased invasiveness and 1223 dysregulated genes. Integration with TCGA-HNSC data identified a Nic-HNC gene set of 168 genes (149 oncogenes, 19 tumor suppressors) as potential biomarkers, including 36 oncogenes overexpressed in heavy smokers. Pathway analysis revealed the upregulation of oncogenic signaling pathways (such as PI3K-AKT) and the suppression of immune responses (such as NF-κB signaling), contributing to tumor aggressiveness. Drug repurposing analysis across GDSC, CTRP, and PRISM databases identified five candidate compounds, with AZD1332 and JAK-8517 showing strong potential to counteract nicotine-driven oncogenes. These findings provide novel insights into nicotine-induced oncogenesis, propose potential therapeutic strategies, and support precision medicine approaches for tobacco-associated HNC.

## 1. Introduction

Head and neck cancer (HNC) ranks among the most prevalent malignancies globally, representing approximately 5% of all cancer cases according to GLOBOCAN 2022 estimates [1]. Standard treatments, including surgery, radiotherapy, and chemotherapy—used alone or in combination—have improved with advances in diagnostic and therapeutic strategies over the past decade. Nevertheless, late-stage HNC prognosis remains poor, driven by high rates of therapeutic resistance and metastasis [2]. Established risk factors for HNC carcinogenesis include human papillomavirus (HPV) infection, excessive alcohol consumption, areca nut chewing, smokeless tobacco use, and cigarette smoking [3,4,5,6,7,8,9,10]. HPV infection predominantly contributes to oropharyngeal cancer [3], while other factors primarily affect the oral cavity [4,5,6,7,8,9,10]. Areca nut chewing is a significant risk factor in Southeast Asia [5,6], whereas tobacco use—encompassing cigarette smoking and smokeless tobacco—stands as a leading global contributor to HNC [7,8,9,10]. Cigarette smoking, involving the inhalation of tobacco combustion byproducts, is also associated with lung cancer and cardiovascular diseases [7]. In contrast, smokeless tobacco, absorbed through the oral mucosa, increases risks of gum disease, oral cancer, and metabolic disorders [8]. Notably, tobacco use correlates strongly with aggressive cancer phenotypes and elevated mortality rates, highlighting its substantial role in HNC progression [9,10].

Tobacco harbors numerous carcinogenic compounds, such as tobacco-specific nitrosamines, polycyclic aromatic hydrocarbons, volatile organic compounds, and nicotine [7,8]. Nicotine, the predominant alkaloid in tobacco, serves as the primary psychoactive agent driving addiction [11]. Beyond its psychological effects, nicotine and its derived nitrosamines promote carcinogenesis by binding to nicotinic acetylcholine receptors (nAChRs), activating oncogenic pathways including MAPK and PI3K/AKT [12,13,14,15]. Furthermore, nicotine-derived nitrosamines exhibit mutagenic and tumor-promoting properties, inducing genomic damage—such as p53 mutations—that facilitate cancer initiation and progression [15,16]. Nicotine also fosters tumorigenic conditions, including metabolic dysregulation, epithelial-to-mesenchymal transition, and the emergence of cancer stem cell-like phenotypes, collectively enhancing tumor growth, therapy resistance, and metastasis [17,18]. Clinical data reinforce these mechanisms, demonstrating reduced efficacy of anticancer therapies and poorer outcomes in tobacco users [19,20,21,22].

Prior studies have identified nicotine-associated oncogenes and their regulatory mechanisms [12,13,14,15,16,17,18,19,20,21,22], while transcriptomic analyses have illuminated broader carcinogenic effects of tobacco exposure. For instance, Wang et al. performed transcriptomic profiling of lung epithelial cells exposed to an IC50 dose of cigarette smoke condensate for 24 h, revealing dysregulated genes linked to cell cycle regulation, DNA repair, cancer, and metabolic pathways [23]. Similarly, Boyle et al. conducted a comparative analysis of oral mucosal tissues from 40 current smokers and 40 never-smokers, identifying 41 differentially expressed genes involved in xenobiotic metabolism, oxidative stress, eicosanoid synthesis, and nicotine signaling [24]. However, these studies show limited gene overlap, likely due to variations in experimental models and methodologies, with acute exposure models capturing short-term responses and pre-malignant tissue analyses missing cancer-specific alterations. Notably, while tobacco’s role in HNC initiation is well established, its contribution to poor prognosis remains poorly elucidated due to a paucity of comprehensive molecular studies [9,10,19,20]. This critical gap highlights the need for a detailed transcriptomic profile of nicotine-induced carcinogenesis in HNC to inform prognostic and therapeutic strategies.

Understanding nicotine-driven molecular alterations is essential for identifying therapeutic targets and designing targeted interventions. To simulate chronic nicotine exposure in habitual tobacco users and capture long-term cellular adaptations, we developed a cellular model by exposing human HNC cell lines to a low-dose (IC30) nicotine concentration for three months. We conducted transcriptomic profiling of these nicotine-adapted sublines and integrated the results with clinically relevant cancer-related alterations from The Cancer Genome Atlas Head and Neck Squamous Cell Carcinoma (TCGA-HNSC) dataset. This integrative analysis delineated a gene panel implicated in tobacco-induced HNC and its role in nicotine-driven tumor progression. Additionally, we employed in silico drug repurposing to identify therapeutic candidates capable of mitigating nicotine-induced oncogenesis, further assessing their predicted efficacy against key genes dysregulated in HNC patients with heavy smoking histories. This study integrates transcriptomic and pharmacological approaches to comprehensively analyze nicotine-driven molecular changes in HNC, and identifies promising therapeutic candidates for tobacco-related malignancies.

## 2. Materials and Methods

### 2.1. Cell Culture and Maintained Nicotine-Treated Sublines

HNC cell lines OECM1, SAS, and CGHNC9 were utilized [25]. OECM1 cells were cultured in RPMI 1640 medium (Thermo Fisher Scientific, Waltham, MA, USA), whereas SAS and CGHNC9 cells were maintained in Dulbecco’s Modified Eagle Medium (DMEM, Thermo Fisher Scientific). All media were supplemented with 10% fetal bovine serum (FBS, Thermo Fisher Scientific) and 1% antibiotic-antimycotic solution (Thermo Fisher Scientific). Cells were maintained under standard conditions in a humidified incubator at 37 °C with 5% CO_2_. To generate nicotine-adapted sublines, cells were seeded at an initial density of 5 × 10^5^ cells per 10 cm dish and allowed to adhere for 12 h. Nicotine (Sigma-Aldrich, St. Louis, MO, USA) treatment was initiated at the IC30 concentration (1.73 mM for OECM1, 0.875 mM for SAS, and 1.256 mM for the CGHNC9 cell lines), which was determined through a trypan blue exclusion assay. Cells were cultured and maintained in cell line-specific medium containing nicotine, with the medium changed every three days (Figure 1a). Upon reaching approximately 90% confluence, cells were passaged, reseeded, and subjected to the same treatment regimen. This cycle was systematically repeated for three months to establish stable nicotine-adapted sublines.

### 2.2. Nicotine Tolerance

To assess the nicotine tolerance of nicotine-treated sublines, a trypan blue exclusion assay was performed to evaluate cell viability. Parental HNC cells and nicotine-treated sublines were seeded at a density of 1 × 10^5^ cells per well in 6-well plates and exposed to a serial dilution of nicotine (0~4 mM) for 72 h. Following treatment, viable cells were counted and normalized to the untreated control group, which was set as 100% viability. This normalization enabled a comparative evaluation of nicotine tolerance between parental and nicotine-treated sublines.

### 2.3. Colony Formation Assay

The long-term growth effects were assessed using a colony formation assay, following previously established protocols [25]. Cells were seeded at 800 cells per well in 3 cm culture dishes and maintained under standard culture conditions (37 °C, 5% CO_2_) for 7–10 days. After incubation, colonies were fixed with formaldehyde (Sigma-Aldrich), stained with 0.5% crystal violet (Sigma-Aldrich) for 30 min, washed with water, and air-dried. The number of colonies was quantified using ImageJ software (version 1.53a; NIH, Bethesda, MD, USA).

### 2.4. Matrigel Invasion Assay

The cell invasion ability was evaluated following previously established protocols [26] using Matrigel-coated Millicell invasion chambers (8 μm pores). HNC cells were seeded in the upper chamber with 1% FBS medium, while the lower chamber contained 10% FBS medium as a chemoattractant. After incubation, non-invading cells were removed, and invaded cells were fixed with formaldehyde, stained with crystal violet, and imaged microscopically. The invaded area was quantified using ImageJ software.

### 2.5. Transcriptomic and Pathway Enrichment Analysis

The transcriptomic analysis was conducted following previously established protocols [27] to characterize the gene expression profile associated with prolonged nicotine exposure in HNC cells. Affymetrix cDNA microarray (GeneChip Human Genome HG-U133A, Thermo Fisher Scientific) was utilized to compare transcriptomic differences between three parental HNC cell lines and their corresponding nicotine-treated sublines. Data preprocessing included batch effect removal and normalization using the robust multiarray averaging (RMA) algorithm in Partek^®^ Genomics Suite (Partek, St. Louis, MO, USA). To determine functional pathways associated with prolonged nicotine exposure, differentially expressed genes (DEGs) from microarray analysis were analyzed using The Database for Annotation, Visualization, and Integrated Discovery (DAVID) and Kyoto Encyclopedia of Genes and Genomes (KEGG) bioinformatics tools (https://david.ncifcrf.gov/, accessed on 8 November 2024). Pathway enrichment analysis was performed based on the KEGG database, and significantly enriched pathways (*p*-value < 0.05) were reported for both upregulated and downregulated genes by bubble plots.

### 2.6. Drug Repurposing by In Silico Bioinformatic Analysis

Drug repurposing was conducted using an in silico bioinformatic pipeline to identify small-molecule compounds effective against nicotine-driven HNC. The corresponding gene expression data are obtained from through the Broad Institute’s Cancer Cell Line Encyclopedia (CCLE) data. Drug response data were obtained from the Genomics of Drug Sensitivity in Cancer (GDSC), Cancer Therapeutics Response Portal (CTRP), and Profiling Relative Inhibition Simultaneously in Mixtures (PRISM) databases [28,29,30], which provide transcriptomic profiles and drug sensitivity metrics (e.g., IC50) for a broad range of small-molecule compounds (288–4517 drugs) across 578–829 cancer cell lines. Raw data were preprocessed to standardize gene expression and drug response formats, addressing variations in cell line annotations. Transcriptomic profiles from The Cancer Genome Atlas Head and Neck Squamous Cell Carcinoma (TCGA-HNSC) dataset, accessed via the UCSC Cancer Genome Browser, were stratified into non-smokers (category 1, no tobacco use) and heavy smokers (category 2, pack-year value ≥ 40) to compare nicotine-associated molecular signatures. Drug sensitivity predictions were performed using the *oncoPredict* R package [31], which utilizes ridge regression models trained on cell line data from the aforementioned pharmacogenomic datasets. The “calcPhenotype” function was applied to estimate drug sensitivity scores (predicted IC50 values) for each sample, facilitating the identification of candidate compounds with enhanced tumor-specific activity. Drugs with lower predicted IC50 values in either patient group were prioritized as potential therapeutic candidates. Statistical comparisons were conducted using Wilcoxon rank-sum tests, and results were visualized using volcano plots.

To assess the potential relevance of top-ranked drugs in modulating oncogenic activity, Pearson correlation analyses were performed between predicted IC50 values of the top four most sensitive compounds and the expression levels of oncogenic driver genes. In addition, a focused correlation analysis was conducted using nicotine-associated oncogenes, previously identified from transcriptomic analyses of nicotine-treated HNC cell lines. The distribution and significance of drug–gene correlations were illustrated using dot plots and heatmaps. All computational analyses were conducted using R software (version 4.4.2).

## 3. Results

### 3.1. Establishment of Chronic Nicotine-Exposed Head and Neck Cancer Cell Sublines

To establish a model of prolonged nicotine exposure, we generated nicotine-treated sublines from the human HNC cell lines OECM1, SAS, and CGHNC9. These cells were continuously exposed to nicotine at their respective IC30 doses for three months, resulting in the successful derivation of nicotine-adapted sublines: OECM1-Nic, SAS-Nic, and CGHNC9-Nic (Figure 1a). To confirm the adaptation of these sublines to chronic nicotine exposure, we evaluated their nicotine tolerance relative to their parental counterparts. As shown in Figure 1b, all nicotine-exposed sublines demonstrated increased resistance to nicotine, validating their successful adaptation. To further characterize these sublines, we assessed their proliferative capacity and invasive potential using colony formation and Matrigel invasion assays. Colony formation assays revealed no significant differences between nicotine-treated sublines and their parental counterparts (Figure 1c), indicating that chronic nicotine exposure does not markedly affect proliferative ability. In contrast, Matrigel invasion assays demonstrated a significant increase in invasive capacity across all nicotine-treated sublines—OECM1-Nic, SAS-Nic, and CGHNC9-Nic—compared to their respective parental cells (OECM1-Pt, SAS-Pt, and CGHNC9-Pt) (Figure 1d). These results suggest that prolonged nicotine exposure enhances the invasive phenotype of HNC cells, potentially contributing to increased aggressiveness and metastatic potential.

**Figure 1 cancers-17-01430-f001:**
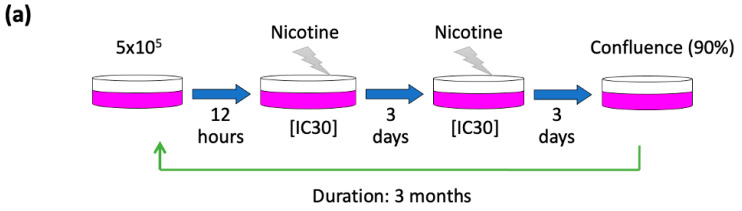
Chronic nicotine treatment enhances the invasion ability in HNC sublines. (**a**) A schematic representation of the protocol for establishing chronic nicotine-treated HNC sublines, as detailed in Section 2. (**b**) Nicotine-treated sublines exhibit increased nicotine tolerance, as assessed by the trypan blue exclusion assay. (**c**) Parental and nicotine-treated sublines show comparable growth rates, as determined by the colony formation assay. Scale bars, 5 mm. (**d**) Nicotine-treated HNC sublines display enhanced invasion ability compared to parental cells, as evaluated by the Matrigel invasion assay. Data are presented as mean ± SD. ***, *p* < 0.001; *n.s.*, not significant. Scale bars, 25 μm.

### 3.2. Global Transcriptomic Alterations Induced by Chronic Nicotine Exposure in HNC Cells

To characterize the molecular landscape associated with prolonged nicotine exposure, we performed differential transcriptomic analyses on nicotine-treated and untreated HNC cell lines using Affymetrix GeneChip microarray technology. As shown in Figure 2a, global transcriptomic profiles of DEGs in nicotine-treated sublines displayed distinct patterns compared to their parental counterparts.

Using a threshold of |FC| ≥ 1.5, we identified significant transcriptional alterations in nicotine-treated sublines. In OECM1-Nic cells, 2452 DEGs were detected, including 1335 upregulated and 1117 downregulated transcripts relative to parental OECM1-Pt cells. Similarly, SAS-Nic cells exhibited 1497 DEGs, with 811 upregulated and 686 downregulated transcripts. In CGHNC9-Nic cells, 3626 DEGs were observed, comprising 1868 upregulated and 1758 downregulated transcripts compared to their parental counterparts.

To delineate a common molecular signature of chronic nicotine exposure, we selected genes consistently dysregulated across at least two nicotine-treated cell lines, resulting in a set of 799 upregulated and 424 downregulated genes (Figure 2b). An overview of dysregulation levels, including geometric mean (GEOMEAN) and *p*-values for each gene, is presented in Figure 2c. Prominent upregulated genes included *MAN2A1*, *NDFIP1*, and *ERAP2*, while *TCF15*, *HNRNPUL2*, and *CXCL2* were among the most significantly downregulated genes. These results establish a core set of molecular perturbations induced by chronic nicotine exposure, laying the groundwork for further studies on the biological consequences and potential therapeutic targets linked to nicotine-driven carcinogenesis in HNC.

### 3.3. Dysregulated Molecular Pathways in Nicotine-Exposed HNC Cells

To elucidate the biological processes underlying the nicotine-adapted phenotype in HNC cells, we performed functional pathway enrichment analyses on the upregulated and downregulated DEGs using the DAVID bioinformatics tool. Enrichment results for the 799 upregulated genes, presented in Figure 3a, revealed pathways grouped into three primary modules: cancer-associated, immune-related, and homeostasis-related. The cancer-associated module showed significant enrichment in pathways such as PI3K-AKT signaling, cell adhesion mechanisms, stress response, and oncoviral infection, suggesting oncogenic activation in nicotine-exposed cells. The immune-related module included pathways linked to viral and parasitic infections, indicating that nicotine exposure may impair immune surveillance and increase pathogen susceptibility. The homeostasis-related module featured metabolic biosynthesis pathways, pointing to nicotine-induced metabolic alterations that may contribute to tumorigenesis.

Pathway enrichment analysis of the 424 downregulated genes, detailed in Figure 3b, similarly categorized them into these three modules. Within the cancer-associated module, pathways such as NF-κB and TNF signaling—known to suppress malignancy—exhibited significant downregulation. The immune-related module encompassed pathways critical for immune response regulation and pathogen defense, suggesting that nicotine exposure may compromise immune function. The homeostasis-related module highlighted pathways tied to metabolic disorders, implying that nicotine-induced gene suppression disrupts cellular metabolic balance and heightens disease susceptibility.

Collectively, these findings demonstrate that nicotine-induced gene alterations exert widespread effects on cellular functions. The upregulation of oncogenic pathways coupled with the suppression of regulatory networks may destabilize cellular homeostasis, weaken immune defenses, and promote malignant transformation. Such dysregulation likely enhances susceptibility to infections, drives tumor progression, and accelerates cancer development in nicotine-exposed HNC cells.

### 3.4. Nicotine-Driven Gene Signatures Associated with HNC Progression

To identify clinically relevant genes implicated in the malignant transformation of head and neck cancer (HNC), we analyzed differential gene expression profiles between normal tissues (N = 44) and tumor tissues (N = 520) from HNC patients using the TCGA-HNSC dataset. Applying stringent selection criteria (|FC| ≥ 2.0 and FDR-adjusted *p* ≤ 0.05), we identified 2579 DEGs, comprising 1891 overexpressed and 688 underexpressed genes in tumor tissues (Figure 4a). Among these, matrix metalloproteinase (MMP) family members—including MMP1, MMP3, MMP9, and MMP11—exhibited significant upregulation, consistent with their established roles in promoting tumor invasiveness. Conversely, genes critical for maintaining epithelial cell integrity, such as CRNN, MUC21, KRT4, and KRT13, were markedly downregulated. These findings indicate that malignant transformation is characterized by the increased expression of proteolytic enzymes that enhance invasiveness and the decreased expression of structural proteins, leading to compromised cellular integrity.

To investigate nicotine-induced genetic alterations contributing to HNC progression, we integrated genes dysregulated by chronic nicotine exposure with DEGs from the TCGA-HNSC dataset, forming the Nic-HNC gene set. This analysis delineated two distinct panels: (1) an oncogenic panel of 149 genes, upregulated by nicotine and overexpressed in tumors, including SERPINE1, MMP13, INHBA, and SPARC (Figure 4b), which may drive nicotine-induced carcinogenesis; (2) a tumor-suppressive panel of 19 genes, downregulated by nicotine and underexpressed in tumors, such as ATP6V0A4, where reduced expression may impair tumor-suppressive activity due to nicotine exposure (Figure 4c). Together, these results define a Nic-HNC dataset of 168 genes (149 oncogenes and 19 tumor suppressors), providing insights into the molecular mechanisms underlying nicotine-driven carcinogenesis. Detailed gene information, including nicotine-induced fold changes and differential expression between normal and tumor tissues, is presented in Appendix A.

To assess the clinical significance of the 149 oncogenes identified in the Nic-HNC gene set, their expression levels were examined in tumor samples from HNC patients with no smoking history (N = 111) and those with a heavy smoking history (N = 82) using the TCGA-HNSC dataset. Applying a *p*-value threshold of ≤0.05, we identified 36 genes with significantly higher expression in heavy smokers (Figure 4d, Table 1). Notable examples include DHCR7 (*p* < 0.0001), ANO1 (*p* = 0.0107), BAG2 (*p* = 0.0003), MMP2 (*p* = 0.0041), DSG2 (*p* = 0.0247), and TRAM2 (*p* = 0.0002) (Figure 4e), underscoring their potential roles in nicotine-driven HNC progression. These findings delineate a molecular signature of nicotine-induced oncogenesis, emphasizing its contribution to aggressive tumor behavior in heavy smokers.

### 3.5. Drug Repurposing for Targeting Nicotine-Driven HNC: Identification of Potential Therapeutics

Given the elevated therapeutic resistance and poorer survival rates observed in smoking patients [19,20,21,22], we investigated potential therapeutic agents targeting nicotine-regulated cancer progression through a drug repurposing strategy (Figure 5a). This approach aimed to identify drugs with selective sensitivity to HNC tumor tissues by analyzing correlations between drug efficacy and gene expression profiles in tumors from heavily smoking patients (N = 82) versus non-smoking patients (N = 111) using the TCGA-HNSC dataset. Transcriptomic profiles were integrated with drug sensitivity data (IC50 values) from three major databases—GDSC, CTRP, and PRISM [28,29,30]—using the *oncoPredict* R package [31]. Applying a linear ridge regression model with thresholds of *p* < 0.1 for GDSC and *p* < 0.05 for CTRP and PRISM, we identified 12 drugs from GDSC, 50 from CTRP, and 321 from PRISM exhibiting high potential sensitivity in HNC patients with a smoking history (Figure 5a,b).

To refine this candidate list, we selected the four most significant compounds from each database for further evaluation: AZD1332, JAK-8517, X123829, and NU7441 from GDSC; BRD-K30748066, neopeltolide, dinaciclib, and GSK-J4 from CTRP; and disulfiram, oncrasin-1, chlorhexidine, and pyrimethamine from PRISM (Figure 5b). We assessed their impact on nicotine-induced HNC by examining correlations between drug IC50 values and expression levels of the 149 oncogenes identified in the Nic-HNC profile. Negative correlations indicate that nicotine-induced gene expression changes enhance drug sensitivity. The analysis revealed specific genes strongly negatively correlated with each compound: AZD1332 (133 genes), JAK-8517 (123 genes), X123829 (72 genes), NU7441 (108 genes), BRD-K30748066 (108 genes), neopeltolide (115 genes), dinaciclib (55 genes), GSK-J4 (57 genes), disulfiram (90 genes), oncrasin-1 (76 genes), chlorhexidine (25 genes), and pyrimethamine (55 genes) (Figure 5c). Notably, five compounds—AZD1332, JAK-8517, NU7441, BRD-K30748066, and neopeltolide—targeted more than two-thirds (≥99 of 149) of these genes, suggesting robust potential to counteract nicotine-induced malignancy (Figure 5b,c). These findings provide compelling evidence for the therapeutic potential of these compounds in mitigating nicotine-driven oncogenic effects.

To further evaluate these candidates in the context of HNC patients with heavy smoking histories, we analyzed correlations between the 36 nicotine-associated oncogenes identified in such patients and gene expression profiles modulated by each drug using the *oncoPredict* R algorithm. Applying a *p* < 0.05 threshold, AZD1332 and JAK-8517 exhibited the strongest inverse correlations, affecting 31 and 28 of these 36 genes, respectively (Figure 5d). The gene panels modulated by these drugs, presented in Figure 5e, include shared targets such as *F2R*, *SPARC*, *MMP2*, and *TRAM2*, indicating their potential as predictive biomarkers for drug sensitivity. These results position AZD1332 and JAK-8517 as promising therapeutic candidates with specific efficacy against nicotine-induced HNC, supporting their potential for clinical application in this context.

## 4. Discussion

Head and neck cancer (HNC) ranks among the most prevalent malignancies globally, with chronic nicotine exposure from tobacco use identified as a primary driver of HNC carcinogenesis. This study adopted a systematic approach to elucidate and address nicotine-induced HNC progression. We established a cellular model of prolonged nicotine exposure using HNC cell lines, demonstrating significant increases in cellular invasiveness without notable changes in proliferation. Transcriptomic profiling revealed 1223 dysregulated genes (799 upregulated and 424 downregulated), and integration with the TCGA-HNSC dataset delineated a Nic-HNC gene set of 168 genes, comprising 149 oncogenes and 19 tumor suppressors linked to HNC progression. Of these, 36 oncogenes exhibited elevated expression in heavy smokers, suggesting their potential as biomarkers for aggressive HNC. Pathway analyses identified nicotine-driven upregulation of oncogenic cascades, such as PI3K-AKT signaling, alongside suppression of regulatory pathways, including NF-κB signaling. Furthermore, drug repurposing analysis identified five potential therapeutic candidates—AZD1332, JAK-8517, NU7441, BRD-K30748066, and neopeltolide—with the first two emerging as the most promising due to their strong inverse correlations with nicotine-induced oncogenes, particularly in heavy smokers. These findings provide a comprehensive molecular framework for nicotine-driven oncogenesis and offer novel therapeutic insights for tobacco-associated HNC.

This study advances prior research by establishing a long-term nicotine exposure model in HNC cell lines (Figure 1a), successfully validated by the sublines’ increased resistance to nicotine treatment (Figure 1b), which mirrors chronic tobacco use in HNC patients. In contrast to Wang et al.’s 24 h cigarette smoke condensate exposure in lung epithelial cells [23], which identified genes tied to cell cycle regulation and DNA repair, our 3-month exposure captures sustained cellular adaptations reflective of habitual tobacco exposure. Similarly, Boyle et al.’s analysis of oral mucosal tissues from smokers versus non-smokers [24] revealed genes associated with xenobiotic metabolism and oxidative stress, showing limited overlap with our Nic-HNC gene set, likely due to differences in tissue state (pre-malignant versus cancerous). A key finding from our subline characterization is that chronic nicotine exposure significantly enhances cell invasion (Figure 1d), with minimal impact on proliferation (Figure 1c). This suggests that nicotine may drive a more aggressive cancer phenotype, potentially contributing to metastasis and poorer clinical outcomes in smokers, as observed in tobacco users [9,10,19,20,21,22]. This cellular result aligns with our transcriptomic identification of upregulated MMP family genes—including *MMP1*, *MMP2*, and *MMP13* (Figure 4a)—consistent with their established roles in invasion-related mechanisms [32]. These results reinforce the robustness of our findings and underscore the distinct contribution of our chronic exposure model in elucidating nicotine-driven HNC progression.

This study comprehensively profiled global transcriptomic alterations in nicotine-exposed HNC cells, identifying 799 upregulated oncogenes and 424 downregulated tumor suppressors (Figure 2b,c). These molecular profiles enabled a systematic analysis of functional pathways dysregulated by chronic nicotine exposure (Figure 3a,b), providing insights into the mechanisms driving enhanced tumor aggressiveness. Upregulation of oncogenic pathways, notably PI3K-AKT and MAPK signaling (Figure 3a), aligns with their established roles in promoting aggressive cancer phenotypes [33,34] and corroborates prior reports of nicotine-induced activation in patient-derived models [13,14]. Conversely, downregulation of NF-κB and TNF signaling pathways (Figure 3b)—typically involved in malignancy suppression during early carcinogenesis [35,36,37]—may diminish cellular defenses against tumorigenesis, thereby exacerbating cancer progression. Additionally, the enrichment of immune-related pathways associated with viral infections (Figure 3a), alongside the suppression of immune response regulation (Figure 3b), suggests that nicotine impairs immune surveillance, potentially heightening susceptibility to oncoviral infections such as HPV (Figure 3a), a recognized HNC risk factor [3]. Metabolic dysregulation, evidenced by altered biosynthesis and disorder-related pathways (Figure 3a,b), likely fosters a tumorigenic microenvironment conducive to epithelial-to-mesenchymal transition and cancer stem cell-like phenotypes, consistent with previous studies [17,18]. These findings illuminate the multifaceted molecular mechanisms through which nicotine enhances HNC aggressiveness, offering a critical foundation for understanding its oncogenic role and informing targeted therapeutic strategies.

To investigate nicotine-induced molecular alterations driving HNC progression, we integrated genes dysregulated by chronic nicotine exposure with the TCGA-HNSC dataset, establishing the Nic-HNC gene set. This analysis yielded an oncogenic panel of 149 genes (Figure 4b) and a tumor-suppressive panel of 19 genes (Figure 4c), revealing a pivotal role for these genes in nicotine-driven HNC. Key oncogenes, including *SERPINE1*, *MMP13*, *INHBA*, and *SPARC* (Figure 4b), emerged as central to these effects, consistent with their reported contributions to cancer progression across various malignancies [38,39,40,41]. To evaluate the clinical relevance of the Nic-HNC gene set, we assessed their expression levels in HNC patients with no smoking history versus those with a heavy smoking history, identifying 36 oncogenes significantly overexpressed in heavy smokers (Figure 4d). Notable examples—such as *ANO1*, *BAG2*, *DHCR7*, *MMP2*, and *TRAM2* (Figure 4e)—align with prior studies linking these genes related to cancer proliferation or metastasis in multiple cancer types [42,43,44,45,46]. This dataset, encompassing both established and novel genes, offers potential biomarkers for identifying nicotine-driven HNC cases and stratifying patients for personalized treatment. By connecting molecular alterations and aggressive tumor behavior, this study elucidates the link between nicotine exposure and the poor prognosis observed in HNC patients with smoking history [9,10,19,20,21,22], emphasizing the critical need for targeted interventions to improve therapeutic outcomes in this high-risk population.

Given the elevated therapeutic resistance and poorer survival rates observed in HNC patients with smoking history [19,20,21,22], we explored potential therapeutic agents targeting nicotine-regulated cancer progression through a drug repurposing strategy. By integrating transcriptomic data with drug sensitivity profiles from the GDSC, CTRP, and PRISM databases (Figure 5a), this approach exemplifies a cost-effective method to repurpose existing drugs for tobacco-associated HNC, with potential applicability to identifying therapeutics for other diseases. Our analysis identified five promising candidates—AZD1332, JAK-8517, NU7441, BRD-K30748066, and neopeltolide—each exhibiting strong inverse correlations with over two-thirds of nicotine-induced oncogenes (Figure 5c), suggesting their capacity to mitigate the enhanced aggressiveness and invasiveness driven by chronic nicotine exposure. Notably, AZD1332 and JAK-8517 demonstrated robust inverse correlations with 31 and 28 of the 36 oncogenes overexpressed in HNC patients with heavy smoking histories (Figure 5d), underscoring their specific efficacy against nicotine-driven tumor progression. Shared targets, including *F2R*, *SPARC*, *MMP2*, and *TRAM2* (Figure 5e), may serve as predictive biomarkers for drug sensitivity, facilitating precision medicine approaches. By addressing key drivers of invasiveness and aggressiveness, these compounds offer a novel therapeutic avenue for tobacco-associated HNC. These findings establish a foundation for further pharmacological development, highlighting the potential of this cost-effective strategy to improve outcomes in high-risk HNC populations and inform drug discovery efforts across other disease contexts.

This study yields several critical insights into nicotine-driven head and neck cancer (HNC), offering a foundation for advancing precision medicine, though limitations highlight areas for further exploration. The transcriptomic analysis reveals a Nic-HNC gene set, illuminating dysregulated pathways like MAPK and JAK2/STAT3 that drive aggressive tumor behavior, yet reliance on HNC cell lines may not fully capture the in vivo tumor microenvironment’s complexity, potentially constraining translational impact. Similarly, the gene set’s promise as a biomarker to stratify patients requires validation in larger, diverse cohorts to confirm its utility across HNC subtypes and smoking histories. The in silico drug repurposing analysis further identifies candidates—AZD1332, JAK-8517, NU7441, BRD-K30748066, and neopeltolide—predicted to target key genes such as EGFR and JAK2, providing a novel strategy to counteract nicotine-induced oncogenesis, though their efficacy and safety await experimental confirmation. These insights pave the way for personalized treatments, where the Nic-HNC gene set could guide drug selection, matching agents like AZD1332 or JAK-8517 to patients’ molecular profiles to restore treatment sensitivity. Future research should leverage in vivo models to validate these findings, explore the gene set’s applicability to other tobacco-related cancers like lung cancer, and pursue preclinical trials to evaluate the proposed drugs’ therapeutic potential. Developing combination therapies targeting multiple pathways could further amplify efficacy, building on these insights to improve outcomes for HNC patients with a smoking history.

## 5. Conclusions

This study clarifies nicotine’s role in promoting HNC progression through increased invasiveness, molecular dysregulation, and oncogenic gene activation. The Nic-HNC gene set, alongside therapeutic candidates AZD1332 and JAK-8517, establishes a dual framework for risk stratification and targeted intervention in tobacco-associated HNC. These findings highlight the critical need to address nicotine-driven carcinogenesis and set the stage for precision therapies to enhance outcomes in this formidable malignancy.

## Figures and Tables

**Figure 2 cancers-17-01430-f002:**
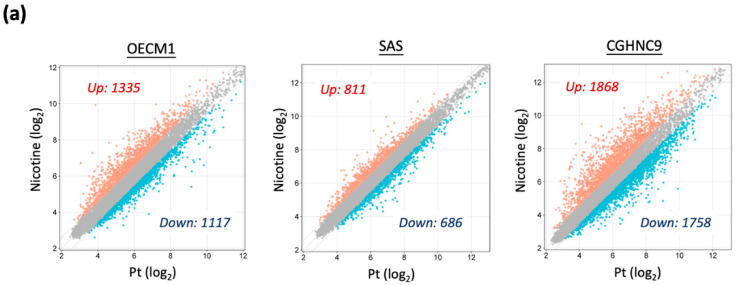
Transcriptomic analysis of dysregulated gene expression in nicotine-treated HNC sublines. (**a**) Transcriptomic profiling revealing differentially expressed genes (DEGs) in nicotine-treated sublines compared to parental cells.The gray dots represent DEGs with no change between the nicotine-treated sublines and the parental cells, while the blue and orange dots represent downregulated and upregulated genes, respectively. (**b**) Commonly dysregulated genes shared across multiple cell lines. (**c**) Volcano plot showing differential gene expression between parental cells and nicotine-treated sublines, with gene dysregulation levels, represented by geometric mean (GEOMEAN) expression and statistical significance (*p*-value) for each gene. Gray dots indicate genes with no change compared to parental cells; blue and red dots indicate downregulated and upregulated genes, respectively. Green dash line indicates GEOMEAN = 1.

**Figure 3 cancers-17-01430-f003:**
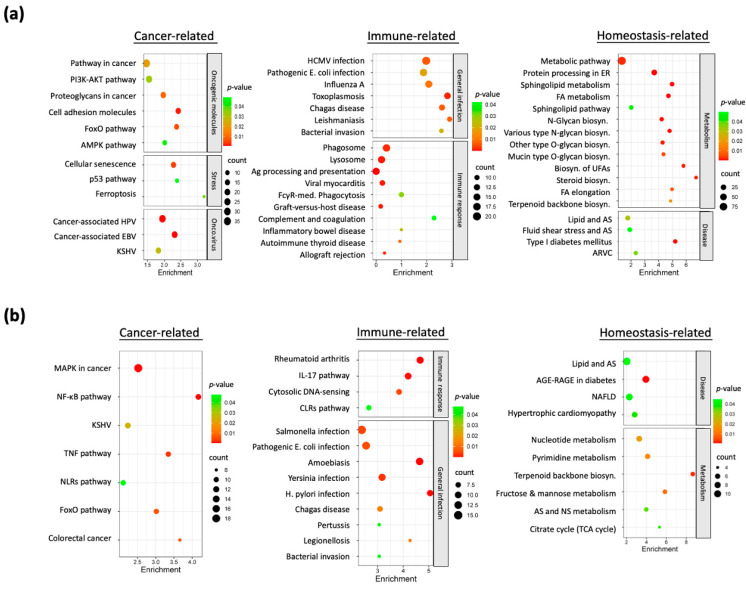
KEGG pathway enrichment analysis of significant genes in nicotine-treated HNC sublines. (**a**) KEGG pathway enrichment analysis of upregulated genes, categorized into distinct functional modules. (**b**) KEGG pathway enrichment analysis of downregulated genes, categorized into distinct functional modules.

**Figure 4 cancers-17-01430-f004:**
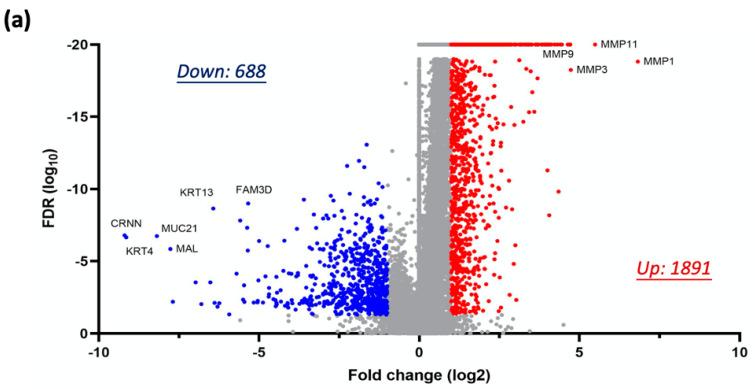
Nicotine-driven gene expression profiles and their association with HNC malignant transformation. (**a**) A volcano plot depicting the differentially expressed genes (DEGs) associated with malignant transformation in the TCGA-HNSC cohort. Gray dots indicate DEGs with no significant change between the tumor and normal groups; blue and red dots indicate underexpressed and overexpressed genes, respectively. (**b**) A Venn diagram illustrating the overlap of upregulated oncogenic genes between the TCGA-HNSC cohort and nicotine-induced gene expression profiles. (**c**) A Venn diagram illustrating the overlap of downregulated tumor-suppressive genes between the TCGA-HNSC cohort and nicotine-reduced gene expression profiles. (**d**) A scatter plot illustrating the ratio of 149 oncogenes expression (Heavy-smoker/Non-smoker) in the TCGA-HNSC cohort, highlighting nicotine-associated oncogenes (red dots). (**e**) Dot plots showing the expression levels of six nicotine-associated oncogenes (DHCR7, ANO1, BAG2, MMP2, DSG2, and TRAM2) in non-smoker (Non) and heavy-smoker (Heavy) HNC patients. Statistical significance was determined using the Wilcoxon rank-sum test.

**Figure 5 cancers-17-01430-f005:**
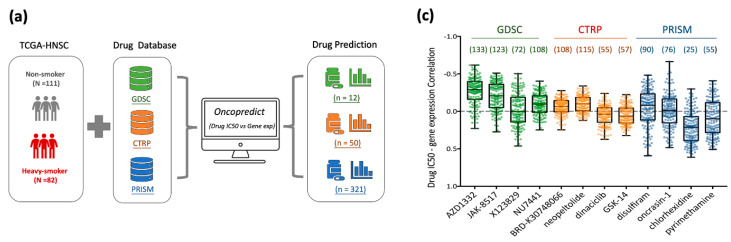
Drug repurposing for targeting nicotine-driven HNC. (**a**) A schematic representation of the drug screening workflow for nicotine-associated HNC using the oncoPredict R package. (**b**) Upper panel: The analysis of drug sensitivity in TCGA-HNSC datasets, including 111 non-smoking patients and 82 heavy-smoking patients. Sensitivity differences in estimated IC50 values were predicted using PRISM, GDSC, and CTRP as training datasets. The *X*-axis represents the sensitivity difference in estimated IC50 values, while the *Y*-axis shows the corresponding *p*-value. Gray dots represent drugs that did not pass the statistical thresholds (*p* < 0.1 for GDSC and *p* < 0.05 for CTRP and PRISM) when comparing the heavy-smoking group to the non-smoking group. Drugs showing increased sensitivity in the heavy-smoking group are marked in red, while those with no increased sensitivity are marked in green, within the statistical thresholds. The horizontal pink dashed lines indicate statistical significance cutoffs (*p* < 0.1 for GDSC; *p* < 0.05 for CTRP and PRISM), and the vertical pink dashed line represents a sensitivity difference of zero. Lower panel: The predicted results of the top four most sensitive drugs in non-smoking and heavily smoking HNC patients based on data from the GDSC, CTRP, and PRISM databases (*, *p* < 0.05; **, *p* < 0.01; ***, *p* < 0.001). (**c**) Boxplots displaying pairwise Pearson correlation coefficients between IC50 values of the top four drugs (from GDSC, CTRP, and PRISM databases) and 149 oncogenic genes in combined datasets. (**d**) A dot plot illustrating the distribution of correlation coefficients between five potential therapeutic compounds and 36 nicotine-associated oncogenes. The *X*-axis represents the correlation coefficients, while the *Y*-axis indicates the corresponding *p*-value. The pink dashed line represents *p* < 0.05. Red dots indicate the 36 nicotine-associated oncogenes. (**e**) A heatmap depicting changes in correlation coefficients of the 36 nicotine-associated genes in response to treatment with five selected drugs. The color scale reflects the correlation coefficient changes.

**Table 1 cancers-17-01430-t001:** List of 36 nicotine-associated oncogenes.

Gene Symbol	Non-Smoker Average (TPM)	Heavy Smoker Average (TPM)	Heavy/Non-Smoker (Ratio)	Wilcoxon Test (*p*)
*ACVR1*	19.8436	24.6753	1.2435	0.0042
*ANO1*	76.7027	144.7553	1.8872	0.0107
*ATP2C1*	66.6058	77.0756	1.1572	0.0237
*BAG2*	2.0117	3.1045	1.5432	0.0003
*CASK*	20.5336	28.0910	1.3681	0.0067
*CHST7*	3.2940	6.0847	1.8472	0.0001
*CNTN1*	7.9286	13.5521	1.7093	0.0008
*DHCR7*	45.8835	90.5598	1.9737	<0.0001
*DSG2*	35.3876	49.0542	1.3862	0.0247
*EPB41L4B*	6.8675	7.7425	1.1274	0.0474
*F2R*	16.2279	19.5841	1.2068	0.0138
*FADS1*	11.6135	16.6838	1.4366	0.0003
*FKBP14*	7.2604	8.5677	1.1801	0.0222
*FOXF2*	4.3628	5.5117	1.2633	0.0096
*HLA-G*	2.4634	10.3195	4.1891	0.0035
*HLTF*	11.4617	12.0126	1.0481	0.0039
*HSPA13*	8.3831	10.0846	1.2030	0.0091
*LAPTM4B*	79.5637	106.3585	1.3368	<0.0001
*LARP6*	11.9819	14.3860	1.2006	0.0276
*LMAN2L*	13.0224	15.2664	1.1723	0.0011
*LRP12*	9.2280	14.1074	1.5288	0.0006
*LRRC8D*	21.0277	26.0205	1.2374	0.0194
*MLF1*	20.5659	32.9833	1.6038	<0.0001
*MMP2*	156.0294	214.7444	1.3763	0.0041
*MYO10*	23.2765	25.2124	1.0832	0.0480
*PLPP2*	59.7515	77.8394	1.3027	0.0003
*PPT1*	63.3588	87.7912	1.3856	0.0164
*RPN1*	202.3942	226.2346	1.1178	0.0052
*SEL1L3*	20.1116	23.9188	1.1893	0.0196
*SLC39A14*	22.6472	27.4927	1.2140	0.0210
*SPARC*	1358.7691	1798.5699	1.3237	0.0111
*SQLE*	68.2900	76.8158	1.1248	0.0299
*STEAP1*	19.8731	22.1430	1.1142	0.0003
*TRAM2*	9.3543	12.5970	1.3466	0.0002
*TUSC3*	32.7518	41.8073	1.2765	0.0089
*UBXN7*	8.9994	11.0759	1.2307	0.0092

## Data Availability

TCGA-HNSC dataset: The gene expression data were obtained from the UCSC Xena platform (https://xena.ucsc.edu/, accessed on 10 May 2024). GDSC drug response data and gene expression data: The dataset was retrieved from the CancerRxGene database (https://www.cancerrxgene.org/downloads/bulk_download, accessed on 10 February 2025). CTRP drug response data: These data were obtained from the Cancer Target Discovery and Development Network, established by the National Cancer Institute’s Office of Cancer Genomics (https://www.cancer.gov/ccg/research/functional-genomics/ctd2/data-portal, accessed on 10 February 2025). The corresponding gene expression data are available through the Broad Institute’s Cancer Cell Line Encyclopedia (CCLE) data portal (https://portals.broadinstitute.org/ccle/data, accessed on 10 February 2025). PRISM drug response and gene expression data: These data were obtained from the depmap R package (Available online: https://bioconductor.org/packages/release/data/experiment/html/depmap.html, accessed on 14 February 2025).

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
