# Peer review of "Deciphering Nicotine-Driven Oncogenesis in Head and Neck Cancer: Integrative Transcriptomics and Drug Repurposing Insights"

_cancers, 2025, doi:10.3390/cancers17091430_

Round 1
Reviewer 1 Report
Comments and Suggestions for Authors
This is an excellent study and a very well-written manuscript. Congratulations to the authors. All experiments were appropriately designed and well presented. The authors are well aware of the limitations of this study. Furthermore, HPV-negative HNCs represent a significant and problematic population of HNCs with unmet medical needs.
-This research reveals that chronic nicotine exposure significantly enhances cell invasion while having minimal impact on cell proliferation in the head and neck cell lines studied. This indicates that nicotine may promote a more aggressive cancer phenotype, potentially leading to metastasis and poorer prognosis and clinical outcomes in smokers.
-The topic is highly relevant to the field and addresses existing gaps. This study advances prior research by establishing a long-term nicotine exposure model in head and neck cancer cell lines. The model's validity is demonstrated by the sublines increased resistance to nicotine treatment, reflecting chronic tobacco use in HNC patients. Additionally, this study contrasts with previous research by Wang et al. [Ref. 23] and Boyle et al. [Ref. 24].
-The authors acknowledge the limitations of their work, as discussed in section 4, and as below:
While this study provides a strong basis for understanding nicotine-driven head and neck cancer (HNC), there are several limitations to consider. Using HNC cell lines instead of primary tissues may not fully capture the complexity of in vivo tumor microenvironments, which could limit the applicability of their findings. Additionally, the clinical relevance of the Nic-HNC gene set needs to be validated in larger and more diverse patient groups to ensure its generalizability across different HNC subtypes and smoking histories.
Future research should prioritize in vivo models to validate these findings, explore the relevance of the Nic-HNC gene set in other tobacco-related cancers like lung cancer, and conduct preclinical trials to assess the therapeutic potential of the proposed drugs.
I have no further comments on this for the authors.
• Are the conclusions consistent with the evidence and arguments presented and do they address the main question posed? Please also explain why this is/is not the case.
Yes, the conclusions consistent with the evidence and argument presented. Furthermore, the conclusions address the main question posed.
This research elucidates how nicotine contributes to the advancement of head and neck cancer (HNC) by enhancing invasiveness, disrupting molecular processes, and activating cancer-causing genes. The Nic-HNC gene set, together with potential treatments AZD1332 and JAK-8517, provides a dual approach for assessing risk and targeting interventions in tobacco-related HNC. These results underscore the urgent need to tackle nicotine-induced cancer development and pave the way for precision therapies to improve outcomes in this challenging disease.
Reviewer 2 Report
Comments and Suggestions for Authors
The manuscript titled “Deciphering Nicotine-Driven Oncogenesis in Head and Neck Cancer: Integrative Transcriptomics and Drug Repurposing In-sights” was reviewed. The present study involves identification of oncogenes overexpressed in heavy smokers as potential biomarkers as well as analysis of five different repurposed drug candidates to counteract nicotine-driven oncogenes. The basic idea of this study is promising; however, the manuscript requires some revisions. The authors are suggested to address following queries.
Detailed methodology should be included in subsection ‘2.6. Drug Repurposing by In Silico Bioinformatic Analysis’.
IC30 and IC50 values / doses should be mentioned.
Some more details and technical insights related to the generated data should be included in the manuscript.
Some sentences are incomplete; should be rechecked and corrected.
Scale bars are missing; should be included in Figure 1.
Explanation / full form of all abbreviations should be included.
Comments on the Quality of English Language
Manuscript should be reviewed to correct typographic/grammatic errors
Reviewer 3 Report
Comments and Suggestions for Authors
The manuscript is very well written and dwells on a very important topic in the understanding of a very important factor influencing prognosis of Head & Neck Cancer.A few minor points need to be addressed as follows:
1.Minor written english language correction is required and a few of them have been highlighted in the attached document.
2.The authors in the Introduction section can add a few lines on the current research gaps in the understanding of role of tobacco in prognosis of Head & Neck Cancer.
3.A few sentences are incoherent and can be rephrased for a better understanding.

The manuscript is very well written and dwells on a very important topic in the understanding of a very important factor influencing prognosis of Head & Neck Cancer.A few minor points need to be addressed as follows:
1.Minor written english language correction is required and a few of them have been highlighted in the attached document.
2.The authors in the Introduction section can add a few lines on the current research gaps in the understanding of role of tobacco in prognosis of Head & Neck Cancer.
3.A few sentences are incoherent and can be rephrased for a better understanding.
Round 2
Reviewer 2 Report
Comments and Suggestions for Authors
The revised manuscript may be considered for publication